# Antibacterial Activity of an FtsZ Inhibitor Celastrol and Its Synergistic Effect with Vancomycin against Enterococci *In Vitro* and *In Vivo*

Xi Lu,[a] Yanxiang Wang,[a] Wei Guo,[a] Zhimeng Zhang,[a] Xinxin Hu,[a] Tongying Nie,[a] Xinyi Yang,[a] Congran Li,[a] Xiukun Wang,[a] Xue Li,[a] Yun Lu,[a] Guoqing Li,[a] ⬤Youwen Zhang,[a] ⬤Lang Sun,[a] ⬤Jing Pang,[a] Xuefu You[a]

[a]Beijing Key Laboratory of Antimicrobial Agents, Institute of Medicinal Biotechnology, Chinese Academy of Medical Sciences and Peking Union Medical College, Beijing, People's Republic of China

Xi Lu and Yanxiang Wang contributed equally to this work. Author order was determined on the basis of contribution.

**ABSTRACT** Enterococci can cause various infectious diseases, including urinary tract infection, wound infection, and life-threatening endocarditis and meningitis. The emergence and transmission of vancomycin-resistant enterococci (VRE) have presented a challenge to clinical treatment. There is an urgent need to develop new strategies to fight against this pathogen. This study investigated the antibacterial and anti-biofilm activity of celastrol (CEL), a natural product originating from *Tripterygium wilfordii Hook F,* against enterococci, and its adjuvant capacity of restoring the susceptibility of VRE to vancomycin *in vitro* and *in vivo*. CEL inhibited all enterococcus strains tested, with MICs ranging from 0.5 to 4 $\mu$g/mL. More than 50% of biofilm was eliminated by CEL at 16 $\mu$g/mL after 24 h of exposure. The combination of CEL and vancomycin showed a synergistic effect against all 23 strains tested in checkerboard assays. The combination of sub-MIC levels of CEL and vancomycin showed a synergistic effect in a time-kill assay and exhibited significant protective efficacy in *Galleria mellonella* larval infection model compared with either drug used alone. The underlying mechanisms of CEL were explored by conducting biomolecular binding interactions and an enzyme inhibition assay of CEL on bacterial cell-division protein FtsZ. CEL presented strong binding and suppression ability to FtsZ, with $K_d$ and $IC_{50}$ values of 2.454 $\mu$M and 1.04 $\pm$ 0.17 $\mu$g/mL, respectively. CEL exhibits a significant antibacterial and synergic activity against VRE *in vitro* and *in vivo* and has the potential to be a new antibacterial agent or adjuvant to vancomycin as a therapeutic option in combating VRE.

**IMPORTANCE** The emergence and transmission of VRE pose a significant medical and public health challenge. CEL, well-known for a wide range of biological activities, has not previously been investigated for its synergistic effect with vancomycin against VRE. In the present study, CEL exhibited antibacterial activity against enterococci, including VRE strains, and restored the activity of vancomycin against VRE *in vitro* and *in vivo*. Hence, CEL has the potential to be a new antibacterial adjuvant to vancomycin and could provide a promising therapeutic option in combating VRE.

**KEYWORDS** vancomycin-resistant enterococci, celastrol, antibacterial activity, synergy, FtsZ

Address correspondence to Jing Pang, pangjing.pangjing@163.com, or Xuefu You, xuefuyou@imb.pumc.edu.cn.

The authors declare no conflict of interest.

Enterococci, commensal bacteria mostly residing in human and animal gastrointestinal tract, are known as one of the leading causes of health care-associated infections (1, 2). More than 20 enterococcal species have been identified, among which *Enterococcus faecalis* (*E. faecalis*) and *Enterococcus faecium* (*E. faecium*) are the most clinically relevant species. The most common and important infections caused by them are urinary tract infections, bacteremia, endocarditis, and surgical site infections. Severe enterococcal infections

**FIG 1** Chemical structure of celastrol.

are associated with mortality rates as high as 20 to 40% during the past 3 decades, despite improvements in medical conditions (3). Moreover, some reports have revealed a rise of ampicillin- and vancomycin-resistant enterococcal infections in health care facilities (4). The emergence and dissemination of multidrug-resistant enterococcal pathogens, especially vancomycin-resistant enterococci (VRE), pose a serious threat to health care practices and makes treatment particularly challenging. The World Health Organization considers vancomycin-resistant *E. faecium* a "high priority pathogen" (5). Therefore, there is an urgent demand for novel antibacterial agents or new strategies to overcome infections caused by VRE.

Celastrol (CEL), as shown in Fig. 1, is a principal bioactive ingredient of *Tripterygium wilfordii Hook F* and has attracted extensive attention due to its multiple promising biological activities. CEL is recognized as a star molecule due to its extensive anticancer, anti-inflammatory, anti-obesity, cardioprotective, antimicrobial, antioxidant, anti-allergic, neuroprotective, anti-thrombotic, anti-osteoarthritic, and anti-Alzheimer's disease activities (6). CEL has been reported to display anti-staphylococcal activity against both planktonic cultures and biofilms (7, 8). However, in-depth research focused on the anti-enterococcal activity of CEL has not been seen.

Here, we report a promising synergistic combination of vancomycin and CEL, which has potential for the use in the treatment of VRE infections. In this study, we systematically evaluated the activity of CEL, alone and in combination with vancomycin, against *E. faecalis* and *E. faecium*, especially VRE *in vitro* and in a *Galleria mellonella* larval infection model. In addition, the actions of CEL on bacterial division were examined to investigate its underlying mechanism of action.

## RESULTS

**Antibacterial activity of CEL.** The minimal inhibitory concentrations (MICs) values of CEL and vancomycin against 48 isolates of *E. faecalis* and 52 isolates of *E. faecium* are summarized in Table 1. Twenty three percent (23/100) of the stains, including 4 *E. faecalis* strains and 19 *E. faecium* strains, were resistant to vancomycin. CEL showed antibacterial effect against all of the vancomycin-susceptible and resistant strains, with MICs ranging from 0.5 to 4 $\mu$g/mL. This could indicate that CEL exerts its antibacterial activity by a mechanism of action different from that of vancomycin and is unlikely to develop cross-resistance with vancomycin.

As shown in Fig. 2, CEL exhibited a concentration-dependent antibacterial effect on enterococci. For all strains tested, significant antibacterial activity was noted at 24 h postinoculation at the concentration of 4 MIC, namely, 16, 8, 16, 8, 8, and 8 $\mu$g/mL for *E. faecalis* ATCC 700802, ATCC 51575, 09-9, *E. faecium* ATCC 700221, ATCC 51559 and 8-30, respectively. CEL, at a concentration of 2 MIC for each strain, exhibited bacteriostatic activity for these six strains and lasted to 24 h. For the concentration of MIC, *E. faecalis* ATCC 700802, 09-9 and *E. faecium* 8-30 were still inhibited by CEL at 24 h after incubation, whereas

**TABLE 1** MICs of CEL and vancomycin against standard and clinical isolated enterococci

| Strains | VRE | VAN[a] | CEL | AMP[b] | Strains | VRE | VAN | CEL | AMP |
|---|---|---|---|---|---|---|---|---|---|
| E. faecium ATCC 700221 | + | >256 | 2 | 128 | E. faecalis ATCC 51299 | + | 16 | 2 | 1 |
| E. faecium ATCC 51559 | + | >256 | 2 | 128 | E. faecalis ATCC 51575 | + | >256 | 2 | 2 |
| E. faecium 15-6 | + | >256 | 4 | 256 | E. faecalis ATCC 700802 | + | 16 | 4 | 1 |
| E. faecium 14-7 | + | >256 | 2 | 256 | E. faecalis 09-9 | + | >256 | 4 | 8 |
| E. faecium 14-11 | + | >256 | 4 | >256 | E. faecalis ATCC 29212 | | 4 | 4 | 1 |
| E. faecium 14-12 | + | >256 | 4 | >256 | E. faecalis ATCC 29302 | | 2 | 2 | 1 |
| E. faecium 14-13 | + | >256 | 4 | 256 | E. faecalis ATCC 51188 | | 2 | 1 | 1 |
| E. faecium 14-15 | + | >256 | 4 | >256 | E. faecalis 18-1 | | 1 | 4 | 1 |
| E. faecium 14-16 | + | >256 | 4 | >256 | E. faecalis 18-2 | | 2 | 4 | 1 |
| E. faecium 14-18 | + | >256 | 4 | >256 | E. faecalis 18-3 | | 4 | 4 | 1 |
| E. faecium 14-19 | + | >256 | 4 | >256 | E. faecalis 18-4 | | 1 | 4 | 128 |
| E. faecium 12-1 | + | >256 | 4 | 256 | E. faecalis 16-1 | | 1 | 2 | 256 |
| E. faecium 12-2 | + | >256 | 4 | 256 | E. faecalis 16-3 | | 2 | 2 | 1 |
| E. faecium 11-3 | + | 64 | 2 | 128 | E. faecalis 16-7 | | 4 | 2 | 1 |
| E. faecium 11-4 | + | 128 | 2 | 128 | E. faecalis 16-8 | | 4 | 2 | 0.5 |
| E. faecium 09-21 | + | >256 | 4 | 256 | E. faecalis 15-18 | | 2 | 2 | 1 |
| E. faecium 09-25 | + | >256 | 4 | 256 | E. faecalis 16-11 | | 2 | 2 | 1 |
| E. faecium 08-11 | + | >256 | 4 | 256 | E. faecalis 15-19 | | 2 | 2 | 1 |
| E. faecium 08-30 | + | 256 | 2 | 128 | E. faecalis 16-13 | | 2 | 2 | 1 |
| E. faecium ATCC19434 | | 1 | 2 | 1 | E. faecalis 16-14 | | 2 | 2 | 2 |
| E. faecium ATCC35667 | | 1 | 2 | 2 | E. faecalis 16-15 | | 2 | 2 | 1 |
| E. faecium 16-1 | | 2 | 2 | >256 | E. faecalis 16-16 | | 2 | 1 | 0.5 |
| E. faecium 16-2 | | 1 | 2 | 0.5 | E. faecalis 16-17 | | 2 | 2 | 1 |
| E. faecium 16-3 | | 2 | 2 | 256 | E. faecalis 16-18 | | 2 | 1 | 128 |
| E. faecium 16-4 | | 1 | 4 | 1 | E. faecalis 16-19 | | 4 | 2 | 1 |
| E. faecium 16-5 | | 2 | 2 | 128 | E. faecalis 16-20 | | 4 | 2 | 1 |
| E. faecium 16-6 | | 1 | 2 | >256 | E. faecalis 16-21 | | 2 | 2 | 1 |
| E. faecium 16-7 | | 2 | 2 | 256 | E. faecalis 16-22 | | 2 | 2 | 1 |
| E. faecium 16-8 | | 2 | 2 | >256 | E. faecalis 16-23 | | 2 | 1 | 2 |
| E. faecium 16-9 | | 2 | 2 | 128 | E. faecalis 16-24 | | 2 | 2 | 0.5 |
| E. faecium 16-10 | | 2 | 4 | 1 | E. faecalis 16-25 | | 2 | 2 | 64 |
| E. faecium 16-11 | | 2 | 2 | >256 | E. faecalis 15-1 | | 2 | 1 | 1 |
| E. faecium 16-12 | | 2 | 2 | 256 | E. faecalis 15-2 | | 1 | 1 | 128 |
| E. faecium 16-13 | | 2 | 2 | 256 | E. faecalis 15-3 | | 2 | 1 | 256 |
| E. faecium 16-14 | | 2 | 2 | 128 | E. faecalis 15-4 | | 2 | 1 | 256 |
| E. faecium 16-15 | | 2 | 0.5 | 256 | E. faecalis 15-5 | | 1 | 1 | 256 |
| E. faecium 16-16 | | 2 | 2 | >256 | E. faecalis 15-6 | | 2 | 1 | 1 |
| E. faecium 16-17 | | 1 | 2 | 128 | E. faecalis 15-7 | | 1 | 1 | 128 |
| E. faecium 16-18 | | 2 | 2 | 128 | E. faecalis 15-8 | | 2 | 2 | 1 |
| E. faecium 16-19 | | 2 | 2 | 128 | E. faecalis 15-9 | | 2 | 2 | 1 |
| E. faecium 16-20 | | 2 | 2 | 128 | E. faecalis 15-10 | | 2 | 1 | 1 |
| E. faecium 16-21 | | 1 | 2 | 256 | E. faecalis 15-11 | | 2 | 2 | 1 |
| E. faecium 16-22 | | 1 | 2 | 256 | E. faecalis 15-12 | | 1 | 2 | 0.5 |
| E. faecium 16-23 | | 2 | 2 | 256 | E. faecalis 15-13 | | 2 | 2 | 1 |
| E. faecium 16-24 | | 1 | 2 | 256 | E. faecalis 15-14 | | 1 | 2 | 4 |
| E. faecium 16-25 | | 2 | 2 | 256 | E. faecalis 15-15 | | 2 | 1 | 128 |
| E. faecium 16-26 | | 2 | 2 | >256 | E. faecalis 15-16 | | 2 | 2 | 1 |
| E. faecium 16-27 | | 4 | 2 | 128 | E. faecalis 15-17 | | 4 | 2 | 1 |
| E. faecium 16-28 | | 1 | 2 | 128 | | | | | |
| E. faecium 16-29 | | 1 | 4 | 1 | | | | | |
| E. faecium 16-30 | | 2 | 1 | >256 | | | | | |
| E. faecium 16-31 | | 2 | 2 | >256 | | | | | |

[a]VAN, vancomycin.
[b]AMP, ampicillin.

regrowth was observed in the other three strains. In addition, no obvious antibacterial activity was seen at 1/4 and 1/2 MIC for any strain.

Biofilms are implicated in difficult-to-treat infections, such as endocarditis, osteomyelitis, chronic wound infections, catheter-associated infections, and prosthetic joint infections. Biofilms are also used as a model to study antimicrobial resistance, and therefore the effect of CEL against VRE biofilms was evaluated. The antibiotics commonly used for the treatment of enterococcal infection, ampicillin and vancomycin,

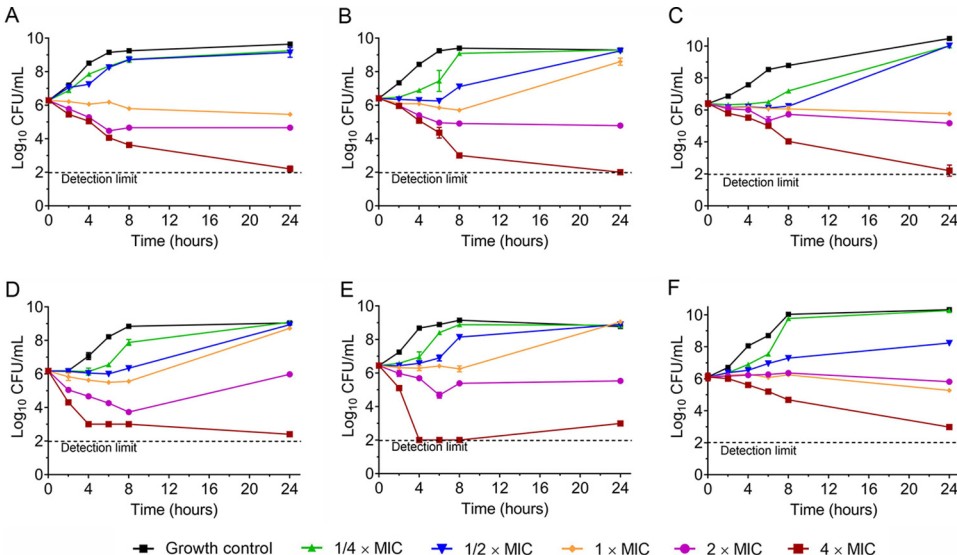

**FIG 2** Activity of CEL at different concentrations in a time-kill analysis against 6 VRE strains. (A) *E. faecalis* ATCC 700802 (MIC = 4 $\mu$g/mL); (B) *E. faecalis* ATCC 51575 (MIC = 2 $\mu$g/mL); (C) *E. faecalis* 09-9 (MIC = 4 $\mu$g/mL); (D) *E. faecium* ATCC 700221 (MIC = 2 $\mu$g/mL); (E) *E. faecium* ATCC 51559 (MIC = 2 $\mu$g/mL); (F) *E. faecium* 08–30 (MIC = 2 $\mu$g/mL).

had little effect on biofilm eradication. In contrast, more than 50% of biofilm of *E. faecalis* ATCC 700802 was eliminated by CEL at 16 $\mu$g/mL after 24 h exposure (Fig. 3).

**Synergistic effect of CEL with vancomycin *in vitro*.** Results of the checkerboard assay are presented in Table 2. Synergistic interactions were observed for the combination of CEL and vancomycin based on the values of fractional inhibitory concentration indices (FICI). The inhibitory concentrations of vancomycin against the strains decreased significantly to concentrations of 1/128 to 1/4 MIC (2 to 16 $\mu$g/mL) when used in combination with CEL at concentrations at a range of 1/8 to 1/4 MIC (0.25 to 1 $\mu$g/mL). According to the FIC indices, the combination of CEL and vancomycin exhibited a synergistic effect in all of the 23 enterococcal strains tested.

The time-kill results of the CEL and vancomycin combination are shown in Fig. 4. At the concentrations of sub-MIC tested, vancomycin used alone exhibited poor inhibitory activity on bacterial growth. CEL used alone showed a weak inhibitory effect on the bacterial growth and regrowth of bacteria was observed at 6 to 24 h postinoculation. The colony-forming units (CFU) counts of individually used vancomycin and CEL were at the same level as growth control of each strain at 24 h. In contrast, the combinations of CEL and vancomycin exhibited a potent synergistic effect on all the 4 tested strains. After 24 h of incubation, the combination led to a significant reduction of viable counts by more than 4log$_{10}$ CFU/mL compared to CEL and vancomycin used separately. In addition, the CEL-vancomycin combination reduced the viable counts by 2log$_{10}$ CFU/mL compared with the starting inoculum. Besides, the antibacterial effects of combinations of CEL at the concentration of MIC and vancomycin at levels of 1/4, 1/2, and 1 MIC were studied with *E. faecalis* ATCC 700802. The result showed that the combinations of CEL and VAN exhibited bacteriostatic activity rather than bactericidal activity even at a relatively higher level of CEL (Fig. S1).

**Synergistic effect of CEL with vancomycin *in vivo*.** *In vivo* experiments were carried out to test whether the combination of CEL-vancomycin could protect larvae of *Galleria mellonella* from a lethal infection dose of the VRE strain *E. faecalis* ATCC 700802 and *E. faecium* ATCC 700221. Larvae were challenged with 2 × 10$^6$ CFU of bacteria and given injections of 10 $\mu$L of PBS, 0.25 mg/kg CEL, 40 mg/kg vancomycin, or CEL-vancomycin combination postinfection. All larvae treated with PBS, CEL, or vancomycin alone became moribund or died within 48 h of VRE inoculation. CEL or vancomycin used alone was insufficient to confer protection. In marked contrast, for larvae infected with *E. faecalis* ATCC

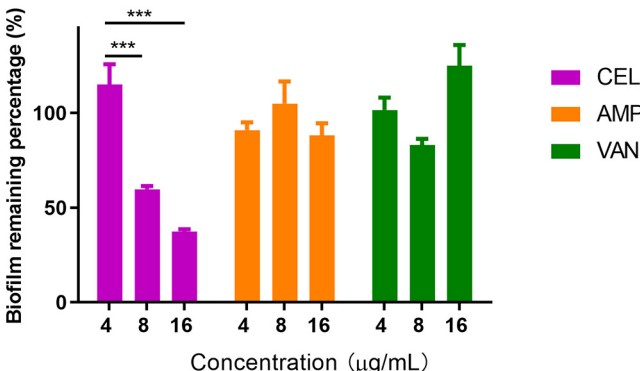

**FIG 3** Concentration-dependent activity on biofilm eradication of CEL, vancomycin, and ampicillin. ($n = 3$, ***, $P < 0.001$).

700802 and *E. faecium* ATCC 700221 and treated with the combination of CEL and vancomycin, the survival percentages are 80% and 90%, respectively, at 96 h postinoculation (Fig. 5). The combination significantly extended survival compared with larvae given PBS or either drug alone ($P < 0.0001$). Thus, the combination of CEL and vancomycin exhibited potent protective efficacy against lethal VRE infection.

**Mechanism of action investigation of CEL.** The underlying mechanism of the antibacterial activity of CEL was first explored by Scanning electron microscopy (SEM) observation of the bacterial cell morphology. As shown in Fig. 6, treatment with CEL did not induce any detectable perturbation of the cell membrane, compared to untreated cells. Rather, CEL treatment caused enlarged enterococcal cells with the average diameters increasing from $1.24 \pm 0.02$ $\mu$m to $1.46 \pm 0.04$ $\mu$m. Besides, significantly elongated cells were observed in the CEL-treated group, and the average total cell length of the dividing bacterial cells increased from $2.01 \pm 0.18$ $\mu$m to $2.65 \pm 0.22$ $\mu$m, which suggests inhibited cell division.

**TABLE 2** Effects of CEL in combination with vancomycin against 23 enterococci strains in checkerboard assay

| Strain | MIC ($\mu$g/mL) alone | | MIC ($\mu$g/mL) combined | | FICI[a] |
|---|---|---|---|---|---|
| | VAN | CEL | VAN | CEL | |
| *E. faecium* ATCC700221 | >256 | 2 | 2 | 1 | 0.50 |
| *E. faecium* ATCC51559 | >256 | 2 | 2 | 0.5 | 0.25 |
| *E. faecium* 15-6 | >256 | 4 | 4 | 1 | 0.26 |
| *E. faecium* 14-7 | >256 | 2 | 4 | 0.5 | 0.26 |
| *E. faecium* 14-11 | >256 | 4 | 2 | 1 | 0.25 |
| *E. faecium* 14-12 | >256 | 4 | 2 | 1 | 0.25 |
| *E. faecium* 14-13 | >256 | 4 | 2 | 1 | 0.25 |
| *E. faecium* 14-15 | >256 | 4 | 2 | 1 | 0.25 |
| *E. faecium* 14-16 | >256 | 4 | 4 | 1 | 0.26 |
| *E. faecium* 14-18 | >256 | 4 | 2 | 1 | 0.25 |
| *E. faecium* 14-19 | >256 | 4 | 2 | 1 | 0.25 |
| *E. faecium* 12-1 | >256 | 4 | 2 | 1 | 0.25 |
| *E. faecium* 12-2 | >256 | 4 | 4 | 1 | 0.26 |
| *E. faecium* 11-3 | 64 | 4 | 2 | 0.5 | 0.16 |
| *E. faecium* 11-4 | 128 | 4 | 2 | 0.5 | 0.14 |
| *E. faecium* 09-21 | >256 | 4 | 2 | 1 | 0.25 |
| *E. faecium* 09-25 | >256 | 4 | 4 | 1 | 0.26 |
| *E. faecium* 08-11 | >256 | 4 | 4 | 1 | 0.26 |
| *E. faecium* 08-30 | 256 | 2 | 2 | 0.5 | 0.26 |
| *E. faecalis* ATCC51299 | 32 | 2 | 8 | 0.25 | 0.38 |
| *E. faecalis* ATCC51575 | >256 | 2 | 8 | 0.25 | 0.14 |
| *E. faecalis* ATCC700802 | 16 | 4 | 2 | 0.25 | 0.19 |
| *E. faecalis* 09-9 | >256 | 4 | 16 | 1 | 0.28 |

[a]For the strains with MICs >256 $\mu$g/mL, the actual FICIs were < the values listed above.

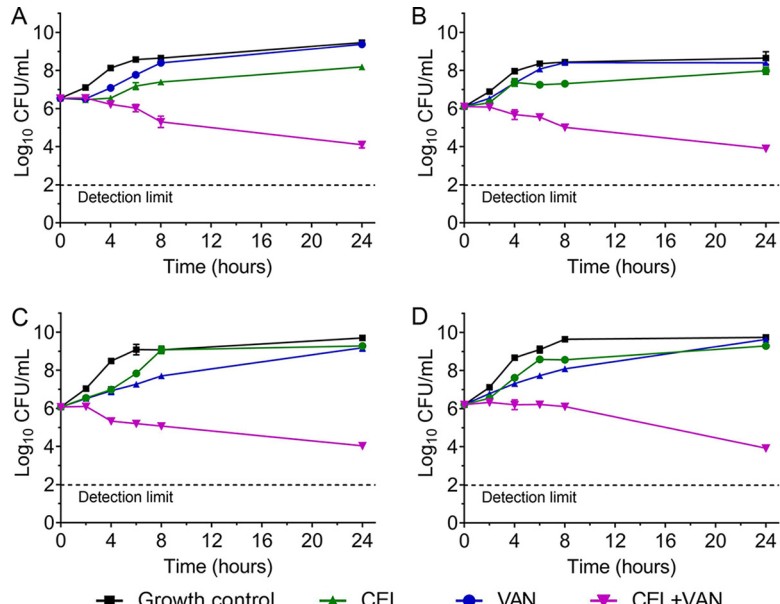

**FIG 4** Sub-MIC levels of CEL and vancomycin alone and in combination against VRE strains. (A) *E. faecalis* 700802, 1/8 MIC (0.5 $\mu$g/mL) CEL and < 1/128 MIC (2 $\mu$g/mL) vancomycin (MIC of CEL = 4 mg/mL, MIC of vancomycin = 16 $\mu$g/mL); (B) *E. faecalis* 09-9, 1/8 MIC (0.5 $\mu$g/mL) CEL and < 1/32 (8 $\mu$g/mL) MIC vancomycin (MIC of CEL = 4 $\mu$g/mL, MIC of vancomycin > 256 $\mu$g/mL); (C) *E. faecium* 700221, 1/4 MIC CEL (0.5 $\mu$g/mL) and < 1/64 MIC (16 $\mu$g/mL) vancomycin (MIC of CEL = 2 mg/mL, MIC of vancomycin > 256 $\mu$g/mL), (D) *E. faecium* 8–30, 1/4 MIC CEL and 1/16 MIC (16 $\mu$g/mL) vancomycin (MIC of CEL = 2 mg/mL, MIC of vancomycin = 256 $\mu$g/mL).

To further confirm our hypothesis, *Bacillus subtilis* (*B. subtilis*) was employed in the cellular morphology investigation. In comparison to the untreated cells, the cell length of *B. subtilis* also significantly increased from 6.28 ± 1.07 $\mu$m to 15.513 ± 4.26 $\mu$m by adding CEL. Given all of this, we speculate that CEL exhibits its antibacterial activity by inhibiting the bacterial cell division.

Filamenting temperature-sensitive mutant Z (FtsZ) is an essential and highly conserved bacterial cytoskeletal cell division protein. Based on the clues we got from SEM and the central role of FtsZ protein in bacterial cell division, we investigated the interaction between CEL and FtsZ to elucidate the antibacterial mechanism of CEL.

Molecular docking was performed between CEL and FtsZ using Discovery Studio 4.5 software. As displayed in Fig. 7, CEL fit well in the active pocket of the FtsZ protein (PDB code: 5MN4), with an ideal docking score of 113.1. Several strong physical interactions,

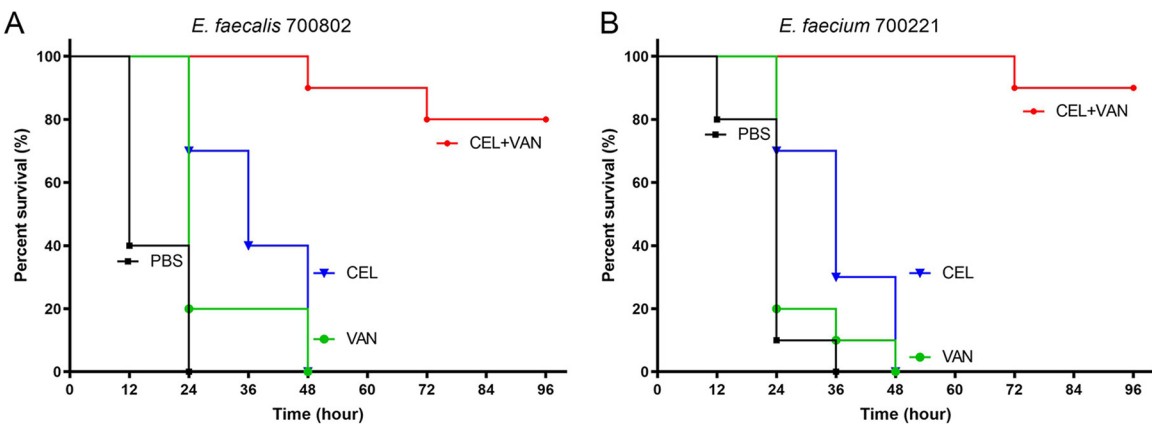

**FIG 5** Protective efficacy of CEL and vancomycin alone and in combination against (A) *E. faecalis* ATCC 700802 and (B) *E. faecium* ATCC 700221. The numbers in parentheses are the numbers of survivors to the total number of larvea challenged. The challenge dose was 2 × 10$^6$ CFU/larvea.

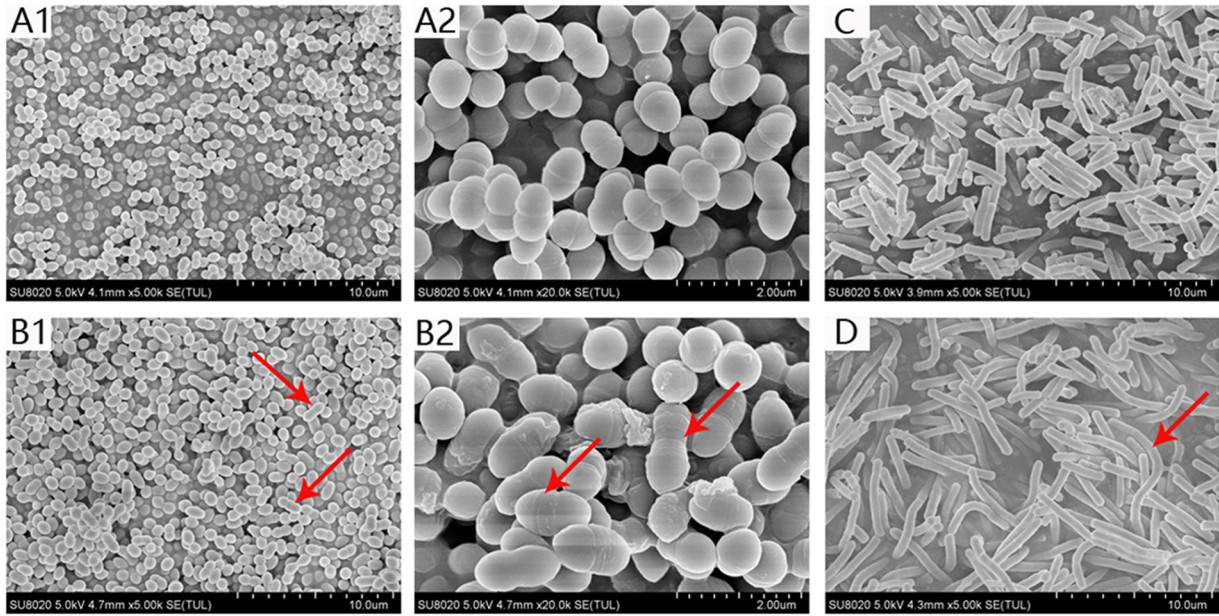

**FIG 6** Images from SEM for *E. faecalis* ATCC 700802 and *B. subtilis* ATCC 21332 after treatments for 4h. (A) *E. faecalis* ATCC 700802 control; (B) *E. faecalis* ATCC 700802 under the treatment of 2 $\mu$g/mL CEL; (C) *B. subtilis* ATCC 21332 control; (D) *B. subtilis* ATCC 21332 treated in 2 $\mu$g/mL CEL.

including one hydrogen bond between oxygen atom of carbonyl (ring A) and ARG143 residue, one hydrogen bond between carboxyl and THR133 residue, and one Pi-anion bond between ring A and GLU139, were predicted. These interactions together contributed to the binding, which suggested that the FtsZ protein might be a potential target of CEL.

To further confirm the interaction strength, the affinity between CEL and FtsZ was evaluated by surface plasmon resonance (SPR) analysis. It was demonstrated that CEL could dose-dependently bind to immobilized FtsZ with a $K_d$ value of 2.454 $\mu$M (Fig. 8), indicating its specific binding ability with FtsZ.

Since the assembly dynamics of FtsZ is considered to be regulated by its GTPase activity, the inhibitory effect of CEL on the GTPase activity of *E. faecalis* FtsZ was evaluated. CEL was found to strongly suppress the GTPase activity of *E. faecalis* FtsZ protein with $IC_{50}$ values of 1.04 $\pm$ 0.17 $\mu$g/mL (Fig. 9). The GTPase inhibitory activity correlates reasonably well with the antimicrobial activity, suggesting that the compound interferes with bacterial growth through a mechanism of FtsZ binding and GTPase function inhibition.

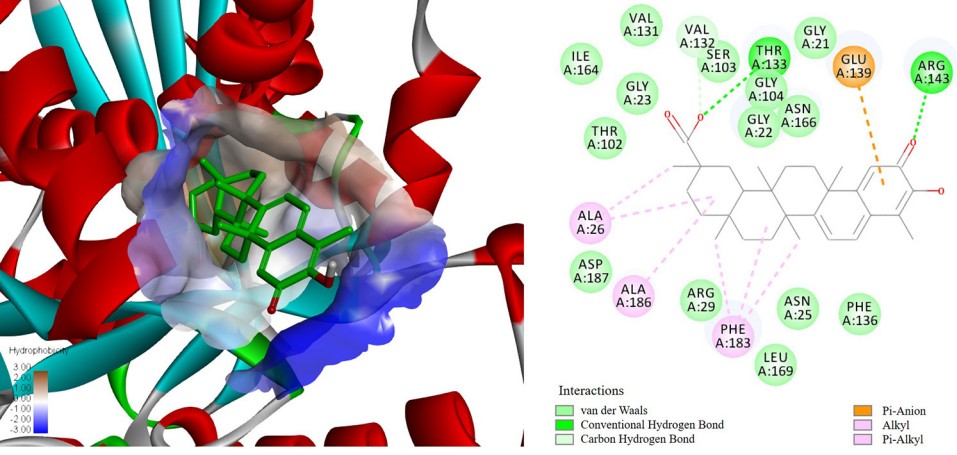

**FIG 7** Docking results of CEL with FtsZ protein, including the interaction of CEL with different amino acids of FtsZ.

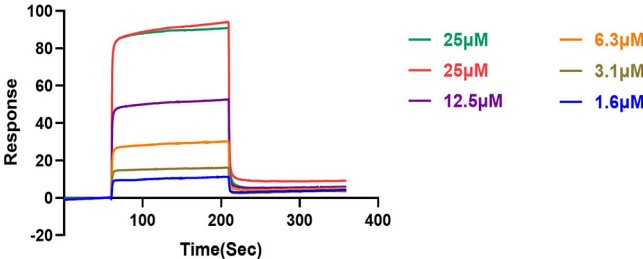

**FIG 8** SPR sensorgrams obtained on an FtsZ-coated chip at different concentrations of CEL.

## DISCUSSION

The rapid emergence of resistant bacteria is occurring worldwide, and it is estimated that the antibiotic resistance would cause 10 million deaths and $100 trillion of economic loss per year by 2050 (9). The emergence, prevalence, and global dissemination of antibiotic resistance are sounding the alarm on the need for new classes of antibacterial agents to counter growing multidrug-resistant pathogenic bacteria. However, novel candidates are increasingly difficult to discover and bring to market, resulting in a growing gap between clinical need and drug innovation. Combinations of antibiotics with nonantibiotic activity-enhancing compounds offer a productive strategy to overcome antibiotic resistance and hamper the progression toward a "postantibiotic" age.

Enterococci, particularly *E. faecalis* and *E. faecium*, are important causes of nosocomial infections and have become a major issue worldwide, causing up to 10% of all infections in hospitalized patients. Moreover, VRE, by acquiring genes through plasmid or transposon that enable bacteria to bypass antibiotic susceptible critical steps in cell wall formation, emerge as a particularly important medical and public health challenge.

CEL, well-known for its activity in treating obesity in recent years (10, 11), exhibits a wide range of biological activities, including significant antibacterial, anti-biofilm, and virulence inhibiting activity against *Staphylococcus aureus* (*S. aureus*). However, the synergistic effect of CEL with vancomycin against VRE has not been previously characterized. In the present study, CEL demonstrated an outstanding antibacterial effect against both vancomycin-susceptible enterococci (VSE) and VRE. The significant synergic effects of CEL to vancomycin were confirmed in the checkerboard assay and the time-kill study *in vitro* and in the *Galleria mellonella* larval lethal infection model *in vivo*.

*Galleria mellonella* larvae have been considered a suitable model for studying human pathogens and increasingly used in antimicrobial efficacy evaluation (12). Considering that the hemolymph accounts for 70% of the larvae weight (13), the dose of CEL (0.25 mg/kg) we used was equivalent to a drug concentration of 0.357 $\mu$g/mL *in vivo*. This *in vivo* dose is close to the *in vitro* MIC of CEL used in combination with vancomycin (0.25 $\mu$g/mL).

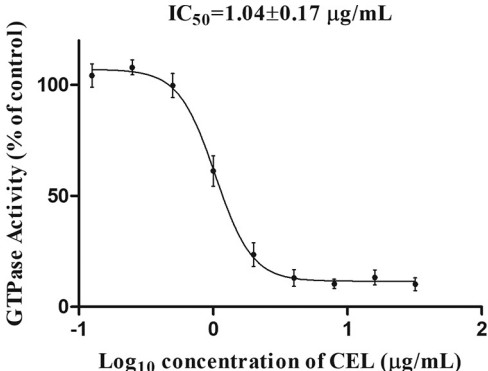

**FIG 9** Concentration-response curve of CEL against the GTPase activity of *E. faecalis* FtsZ. Each point represents the mean of three independent assays.

**TABLE 3** MICs of CEL and vancomycin against *ftsZ* knockdown strain

| MIC ($\mu$g/mL) | WT[a] | *ftsZ* knockdown (xylose induced) | *ftsZ* CRSPRi (uninduced) | *ispH* knockdown (xylose induced) |
|---|---|---|---|---|
| Van | 0.5 | 0.125 | 0.5 | 0.5 |
| CEL | 0.5 | 0.25 | 0.5 | 0.5 |

[a]WT, wild type.

Bacterial cell morphologic change can give valuable clues about the antibacterial mode of action and is frequently used for pilot mechanism investigation. In the CEL-treated group, a Chinese sugar-coated haws on a stick shape for enterococcus and significantly elongated *B. subtilis* were observed, which suggests a cell-division inhibition effect of CEL.

FtsZ is an attractive target for new antibacterial drugs due to its essential role in cell division and high conservation throughout bacteria (14). FtsZ undergoes GTP-dependent polymerization into filaments, which assemble into a highly dynamic cytoskeleton scaffold, known as Z-ring, on the inner membrane of the mid-cell. They then recruit other proteins, which together drive cell division and the formation of new cell poles. Synergistic actions of some FtsZ inhibitors and antibacterial agents were reported in a wide variety of research. The third-generation oral cephalosporin cefdinir and TXA709, an FtsZ-targeting prodrug against methicillin-resistant *S. aureus* (MRSA), exhibited synergy *in vitro* and *in vivo* (15). A quinoline-based FtsZ inhibitor combined with $\beta$-lactam antibiotics shows strong synergistic antimicrobial activities against antibiotic-resistant strains of *S. aureus* (16).

Doxorubicin, a U.S. FDA-approved drug displayed strong interaction with FtsZ (17), was used as a positive control of FtsZ inhibitor in our study. Doxorubicin exhibited synergy with vancomycin, and the MIC of vancomycin against *E. faecalis* ATCC 700802 decreased significantly from 16 $\mu$g/mL to 4 $\mu$g/mL when been used in combination with doxorubicin, with an FIC index of 0.375. Therefore, we assumed that CEL might attack bacteria by inhibiting the function of FtsZ and subsequent biological activity. Therefore, in-depth study was performed to investigate the interaction between CEL and FtsZ, including molecular docking, affinity analysis, and GTPase activity inhibition. All of the results indicated the interaction between CEL and the FtsZ protein.

In order to further verify suppressing GTPase activity of FtsZ as the primary mechanism of CEL, the *B. subtilis ftsZ*-knockdown strain by CRISPR interference (CRISPRi) from *B. subtilis* CRISPRi Essential Gene Knockdown Library (18) was employed in the present study to investigate the effect of vancomycin and CEL on the GTPase-deficient bacteria. Gene knockdown was induced by xylose (1%, freshly prepared). The no sgRNA wild-type strain, an unrelated gene *ispH* knockdown strain, and the uninduced *ftsZ* CRSPRi strain (no xylose added) were used as controls. The MICs of vancomycin and CEL in the noninduced CRISPRi strain and the unrelated gene *ispH* knockdown strain were the same as the no sgRNA wild-type parental strain. As shown in Table 3, lowering the levels of *ftsZ* could increase the susceptibility of vancomycin, with the MIC decreasing from 0.5 to 0.125 $\mu$g/mL. The suppression of FtsZ brought more sensitivity to vancomycin as we expected. Decreased MIC of CEL was also observed in the strain with lower expression level of *ftsZ* compared with the wild-type counterpart, this can be explained by the increased sensitivity of the FtsZ protein in the low expressed strain. While the synergistic effect of CEL and vancomycin could not be seen in *ftsZ* knockdown strain. The possible explanation could be that the deficient GTPase activity (due to gene knockdown) exhibited an equivalent effect produced by a cell division inhibitor (e.g., CEL). Therefore, CEL could not further exhibit reinforcement to the activity of vancomycin in the *ftsZ* knockdown strain, which reconfirmed the action of CEL on the FtsZ protein.

In conclusion, we demonstrated that CEL possessed antibacterial and anti-biofilm activity against enterococci, including VRE strains, and restored the activity of vancomycin against VRE *in vitro* and *in vivo*. Overall, CEL has the potential to be a new antibacterial and antibacterial adjuvant to vancomycin and might provide a new therapeutic option in combating VRE.

## MATERIALS AND METHODS

**Bacterial strains and growth conditions.** All enterococcal bacterial strains used in this study were obtained from the Collection Center of Pathogen Microorganism of Chinese Academy of Medical Sciences (CAMS-CCPM-A) in China, including the American Type Culture Collection (ATCC) reference strains. Five standard VRE strains (2 strains of *E. faecium* and 3 strains of *E. faecalis*), 18 clinical VRE strains (17 strains of *E. faecium* and 1 strain of *E. faecalis*), 5 standard VSE strains (2 strains of *E. faecium* and 3 strains of *E. faecalis*), and 72 clinical VSE isolates (31 strains of *E. faecium* and 41 strains of *E. faecalis*) were included in the current study. *B. subtilis* CRISPRi Essential Gene Knockdown Library was kindly provided by Carol A. Gross (University of California, San Francisco, CA). Strains were routinely grown in cation-adjusted Mueller-Hinton broth (CAMHB) or on Mueller-Hinton agar (MHA) at 37°C.

**Chemicals and agents.** Vancomycin, ampicillin, and CEL were purchased from the National Institutes for Food and Drug Control (Beijing, China). Vancomycin, CEL, and ampicillin were dissolved in distilled water, DMSO, and phosphate buffer (pH 8.0), respectively, as stock solutions at a concentration of 10 mg/mL and stored at −20°C after sterilizing through 0.22 $\mu$m filters. CAMHB, MHA, and Brain Heart Infusion (BHI) used for bacterial culture and antimicrobial susceptibility tests were purchased from Becton, Dickinson and Company (Franklin Lakes, NJ, USA). Doxorubicin and xylose were purchased from Innochem Science & Technology Co., Ltd. (Beijing, China).

CytoPhos phosphate assay Biochem kit and recombinant *E. faecalis* FtsZ protein (FTZ04-B) were purchased from Cytoskeleton (Denver, CO, USA).

**Susceptibility test.** All isolates were stored at −80°C and streaked on MHA plates to make overnight cultures. The MIC of the CEL and vancomycin were determined by the broth microdilution method in accordance with Clinical and Laboratory Standards Institute (CLSI) guidelines (19), with ampicillin as the quality control antibiotic. Compounds with a starting concentration of 256 $\mu$g/mL were added to wells of 96-well microtiter plates and serially diluted with the growth medium CAMHB. A bacterial suspension of 10 $\mu$L was added to reach a final inoculum of $5 \times 10^5$ CFU/mL. The plates were incubated at 37°C for 24 h prior to the reading of MIC. The MIC was defined as the lowest concentration of a compound that inhibited visual growth of the bacteria. All of the MIC values were determined in triplicate on different days. The vancomycin MIC breakpoints are as follows: ≤4 $\mu$g/mL, susceptible; 8 to 16 $\mu$g/mL, intermediate; ≥32 $\mu$g/mL, resistant.

**Time-kill curves of CEL.** Kinetics of antibacterial activity of CEL were assessed by time-kill assay according to the method described by CLSI (20). The experiment was performed on *E. faecalis* ATCC 700802, ATCC 51575, *E. faecium* ATCC 700221, ATCC 51559, and two randomly selected VRE clinical isolates, *E. faecalis* 09-9 and *E. faecium* 08-30. Briefly, overnight cultures were diluted with CAMHB to a final concentration of ~$2 \times 10^6$ CFU/mL and aliquoted to sterile glass tubes, 10 mL per tube. Different concentrations of CEL (1/4 MIC, 1/2 MIC, MIC, 2MIC, and 4MIC) were determined by adding a specific volume of CEL stock solution to each tube. A tube containing culture but no CEL was used as a growth control. The CFU/mL values of the cultures were determined at 0, 2, 4, 6, 8, and 24 h of incubation. Ten microliter samples were taken, 10-fold serial diluted and plated on MHA plates in triplicate, and incubated at 37°C for 24 h. The viable colonies were counted and recorded as $\log_{10}$ CFU per milliliter. Bactericidal activity was defined as ≥ $3\log_{10}$CFU/mL reduction (99.9% kill) in colony count from the starting inoculum. The detection limit was $2\log_{10}$CFU/mL.

**Biofilm killing assay.** Biofilm eradication of the studied strain was performed as previously described (21). *E. faecalis* ATCC 700802 was inoculated into BHI broth and incubated with shaking overnight at 37°C. The 1:100 diluted culture was added to tissue culture-treated 96-well plates, 100 $\mu$L per well, and kept at 37°C under static condition for 24 h to develop biofilm. The medium was gently aspirated, and 100 $\mu$L of fresh medium containing 2-fold diluted CEL or antibiotic control was added (3 replicates). After the incubation of an additional 24 h, the medium was removed and the biofilm was gently washed 3 times with phosphate-buffered saline (PBS) and heat-fixed at 60°C for 1 h. Fifty $\mu$L of 0.06% crystal violet were added to each well and stained for 5 min, and the excess stain was removed by repeated washing with distilled water. The crystal violet was eluted from stained biofilms by adding 200 $\mu$L of 30% acetic acid to each well and transferred to a fresh 96-well plate for OD$_{595}$ reading.

**Studies for synergy.** The combination of CEL and vancomycin was evaluated on 23 VRE strains, including 5 standard strains and 18 randomly selected clinical isolates using a microdilution checkerboard technique (22). Briefly, a final inoculum of $5 \times 10^5$ CFU/mL was added to wells of 96-well microtiter plates containing 2-fold serial diluted CEL and vancomycin in CAMHB. The ranges of CEL and vancomycin dilutions used in the checkerboard titration were 0.125 to 8 $\mu$g/mL and 0.25 to 256 $\mu$g/mL, respectively. After the incubation of 24 h at 37°C, the combined effect of CEL and vancomycin was analyzed by calculating the FIC index using the equation as follows: FICI= (MIC of drug A in the combination/MIC of drug A alone) + (MIC of drug B in the combination/MIC of drug B alone). The combination was defined to be synergistic when the FICI was ≤ 0.5; indifferent when 0.5 ≤ FICI ≤ 4; antagonistic when FICI > 4. The experiments were performed in triplicate on different days.

Time-kill assays were conducted on *E. faecalis* ATCC 700802,09-9 and *E. faecium* ATCC 700221 and 08-30 to evaluate the killing dynamics of CEL and vancomycin. The time-kill study for the combination was similar to that of CEL alone. Compound-free cultures were included for each strain as growth controls. CEL and vancomycin were tested at sub-MICs of each drug. Colony counts were determined at 0, 2, 4, 6, 8, and 24 h. Synergy was defined as ≥ 2 $\log_{10}$ decrease in CFU counts between the combination and its most active constituent at 24 h, plus the number of CFU in the combination at ≥ $2\log_{10}$ CFU/mL below the initial inoculum.

**Infection model of the *Galleria mellonella* larvae.** The larvae were kept at room temperature for 24 h for acclimatization until challenge assay. Larvae with a weight between 270 and 330 mg and a size of

~2 cm were selected. *E. faecalis* ATCC 700802 and *E. faecium* ATCC 700221 were streaked on BHI agar plates, incubated to get colonies, and grown in BHI. Overnight cultures of the two strains were washed and adjusted to the proper density ($\sim2 \times 10^8$ CFU/mL) with PBS and injected into the right posterior proleg of larvae with a glass micro syringe at 10 $\mu$L. Groups of 10 larvae were given vancomycin (40 mg/kg), CEL (0.25 mg/kg), or combination of vancomycin and CEL through the left posterior proleg 1 h postinfection. In addition, 10 larvae were injected with 10 $\mu$L of PBS as negative controls. Larvae were incubated at 35℃ in darkness and monitored for 96 h postinfection. Larvae were scored as dead when they did not respond to a mechanical stimulus. Survival curves were plotted for each group.

**Scanning electron microscopy.** Overnight cultures of *E. faecalis* ATCC 700802 and *B. subtilis* ATCC 21332 were diluted 1:100 in CAMHB and grown to log phase ($OD_{600nm}$ = 0.4) at 37℃. Bacterial cells were then incubated with or without sub-MIC level of CEL for 4 h, collected and fixed with 2.5% glutaraldehyde in PBS for 24 h, rinsed in fresh PBS, passed through an ethanol gradient for dehydration, dried, coated with gold, and observed by scanning electron microscope (Hitachi SU8020, Japan). Cell length was analyzed by ImageJ, an image processing and analysis software available at the NIH website.

**The docking assay.** An automated docking study was carried out using the 3D structure of FtsZ (PDB code: 5MN4) using Discovery Studio 4.5 software. The regularized protein was used in the determination of the important amino acids in the predicted binding pocket. Interactive docking using Libdock protocol was carried out for all the conformers of CEL to the selected active site after energy minimization. The docked compound was assigned a score according to its binding mode onto the binding site.

**Surface plasmon resonance analysis.** SPR assay was performed on a BIAcore T200 biosensor system (GE Healthcare Life Sciences, Piscataway, NJ, USA) at 25℃ using CM5 chip. The binding of FtsZ at different concentrations of CEL (25, 12.5, 6.25, 3.1, 1.6 $\mu$M) was performed in 1×PBS-*P*+ (GE Healthcare Life Sciences) at a flow rate of 20 $\mu$L/min for 120 s. A further dissociation time of 60 s was applied after each injection to allow the signal back to the baseline. $K_d$ value was calculated using Biacore T200 Evaluation software (version 2.0, GE Healthcare Life Sciences, Piscataway, NJ, USA).

**Gtpase activity assay.** The GTPase activity of recombinant *E. faecalis* FtsZ was measured as previously described with minor modifications using a CytoPhos phosphate assay Biochem kit (Cytoskeleton, USA) (23). *E. faecalis* FtsZ (0.5 $\mu$M) was preincubated with vehicle (1% DMSO) or different concentrations of CEL (0.125, 0.25, 0.5, 1, 2, 4, 8, 16, 32 $\mu$g/mL) in 20 mM Tris buffer (pH 7.5) for 10 min at room temperature. Then 5 mM $MgCl_2$ and 200 mM KCl were added. Reactions were started with the addition of 500 $\mu$M GTP and incubated at 37℃. After 30 min, the reactions were quenched by adding 140 $\mu$L of Cytophos reagent and incubated for 10 min. Inorganic phosphate was quantified by measuring the absorbance at 650 nm with a microplate reader (Bio-Rad laboratory Ltd., UK). The relative $IC_{50}$ values were determined by nonlinear regression using a sigmoidal concentration-response curve in GraphPad Prism 8 (GraphPad Software, La Jolla, CA). Three independent assays were performed for all of the tests.

**Statistical analysis.** Statistical significance was determined by one-way analysis of variance (ANOVA) employing SPSS 16.0 and a *P*-value < 0.05 was considered statistically significant.

Survival data were analyzed by Kaplan-Meier survival analysis with a log rank test incorporating Bonferroni's correction for multiple comparisons using GraphPad Prism 8.

## SUPPLEMENTAL MATERIAL

Supplemental material is available online only.
**SUPPLEMENTAL FILE 1**, PDF file, 0.1 MB.

## ACKNOWLEDGMENTS

This research project was supported by grants from National Natural Science Foundation of China (32141003), and CAMS Initiative for Innovative Medicine (2021-1-I2M-030 and 2021-1-I2M-039), People's Republic of China.

We thank Carol Mita (Harvard Medical School) for the help on critical review and revision of the manuscript. We appreciate Carol A. Gross (University of California, San Francisco, CA) for kindly providing the *B. subtilis* CRISPRi Essential Gene Knockdown Library.

Xi Lu, Yanxiang Wang, Jing Pang, and Xuefu You conceived and designed the experiments and wrote the manuscript. Xi Lu, Wei Guo, Zhimeng Zhang, Xinxin Hu, and Tongying Nie conducted the *in vitro* susceptibility experiment. Xiukun Wang and Xue Li assisted with the animal experiment. Yanxiang Wang and Yun Lu conducted the molecular docking and biomolecular binding interaction assay. Youwen Zhang and Lang Sun assisted with SEM characterization. Yanxiang Wang, Xinyi Yang, and Congran Li assisted with revision of the manuscript.

All data were generated in-house, and no paper mill was used. All authors agree to be accountable for all aspects of work ensuring integrity and accuracy.

We have no conflicts of interest to declare.

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
