## [Reviewer comments · Microbiology Spectrum]

Microbiology Spectrum

Antibacterial activity of an FtsZ inhibitor celastrol and its synergistic effect with vancomycin against enterococci in vitro and in vivo

Xi Lu, Yanxiang Wang, Wei Guo, Zhimeng Zhang, Xinxin Hu, Tongying Nie, Xinyi Yang, Congran Li, Xiukun Wang, Xue Li, Yun Lu, Guoqing Li, Youwen Zhang, Lang Sun, Jing Pang, and Xue-Fu You

Corresponding Author(s): Jing Pang, Institute of Medicinal Biotechnology, PUMC

Review Timeline:

Submission Date:	September 13, 2022
Editorial Decision:	October 17, 2022
Revision Received:	November 20, 2022
Editorial Decision:	December 2, 2022
Revision Received:	December 5, 2022
Accepted:	December 9, 2022

Editor: Brian Conlon

Reviewer(s): The reviewers have opted to remain anonymous.

Transaction Report:

DOI: <https://doi.org/10.1128/spectrum.03699-22>

October 17, 2022

Dr. Jing Pang
Institute of Medicinal Biotechnology, Chinese Academy of Medical Sciences & Peking Union Medical College
Beijing
China

Re: Spectrum03699-22 (Antibacterial activity of celastrol and its synergistic effect with vancomycin against enterococci in vitro and in vivo)

Dear Dr. Jing Pang:

Thank you for submitting your manuscript to Microbiology Spectrum. As you will see your paper is very close to acceptance. Please modify the manuscript along the lines recommended by the reviewers. As these revisions are quite minor, I expect that you should be able to turn in the revised paper in less than 30 days, if not sooner. You will find the reviewers' comments below.

When submitting the revised version of your paper, please provide (1) point-by-point responses to the issues raised by the reviewers as file type "Response to Reviewers," not in your cover letter, and (2) a PDF file that indicates the changes from the original submission (by highlighting or underlining the changes) as file type "Marked Up Manuscript - For Review Only". Please use this link to submit your revised manuscript. Detailed instructions on submitting your revised paper are below.

Link Not Available

Sincerely,

Brian Conlon

Reviewer comments:

Reviewer #1 (Comments for the Author):

Summary

The authors have investigated and characterized the mechanism celastrol (CEL) action against enterococci. Their study begins with a series of MIC determinations and checkboard assays, revealing that CEL acts in synergy with vancomycin (VAN). This in itself is an important revelation: the discovery of a molecule that might help in rescuing an antibiotic with reduced clinical utility for VAN-resistant (VRE) strains. Then, through imaging experiments, the authors formulate and test a hypothesis on the MOA. Specifically, observations of atypical cell morphology suggested that FtsZ, an enzyme that contributes to cell division, was important. Remarkably, the authors go on to report values related to FtsZ binding and FtsZ GTPase inhibition that are commensurate with the observed MIC values for CEL (all values are within ~4 fold of each other: ~1-4 ug/mL). Thus, the experimental results provide a compelling argument for a MOA that is dependent on FtsZ inhibition (which could have been included in the title).

Specific comments and suggestions.

1. While I found the experimental results compelling, the writing could be significantly improved. This would not only help readability but also the impact that these exciting results will have with their audience.
2. In general, the use of color in the Figures 2-4 would help the reader.

3. Related to the "time-kill" assay, I would like to note that VAN is generally cited as having a bacteriostatic effect on enterococci and the mechanism of FtsZ inhibition is also suggestive to slow/impair division rather than have an active mechanism of killing (such as membrane disruption). Moreover, the time-dependence of the CFU reduction and the fact that in all cases a substantial number of bacterial cells were recovered after 24 h, the use of the term bacteriocidal may be an overstatement. There does not appear to be a rapid mode of action but rather, I anticipate, a multifactorial process that ultimately leads to cell death. While the authors define bacteriocidal as a reduction by $3\log_{10}$ units (99.9% reduction) the term is a bit confounding (in my opinion).

For the combination treatments CEL + VAN, I would also recommend repeating at least one assay (with one-two strains a VSE and VRE, perhaps) at concentrations closer to the MIC values. This would presumably allow readers to observe a more significant reduction in cell viability over time and better corroborate the aforementioned bacteriocidal claims in the manuscript.

4. The reported FICI are limited by the VAN MIC data. For strains with MICs >256 $\mu\text{g}/\text{mL}$, I do not expect the authors to determine actual values; however, the reported FICI should reflect this. In fact, many of the FICIs are lower than the reported value (e.g. for ATCC51559 the FICI is <0.25). I recommend noting this in the table near the FICI heading.

5. The authors should relate the concentrations used in the Galleria model to those obtained from in vitro MIC determinations.

Reviewer #2 (Comments for the Author):

The manuscript by Lu et al. examines the synergistic effect of celastrol with vancomycin against *Enterococcus faecalis* and *Enterococcus faecium*. Enterococci are serious health concern and the emergence of vancomycin resistant strains highlights the need for new treatment strategies. The research presented here shows that celastrol (CEL) displays anti-biofilm activity and synergizes with vancomycin to kill Enterococci including vancomycin-resistant strains (VRE) in vitro. Additionally, using a *Galleria melonella* larvae infection model, Liu et al. show that the combination of CEL and vancomycin protected larvae from lethal infection of VRE compared to either compound alone. Lui et al. observed elongated cells in SEM micrographs and hypothesized that the filamenting temperature-sensitive mutant Z (FtsZ) might be the target of CEL. They demonstrated by surface plasmon resonance analysis the CEL does have a strong binding ability with FtsZ and suppressed the GTPase activity that is essential for regulating cell division. Overall, the work presented here is very solid. A major highlight is the exciting finding that CEL can render VRE strains sensitive to vancomycin both in-vitro and in-vivo.

Minor comments:

1. The impact of the work shown is somewhat diminished by the fact that the same phenomena reported here has previously been reported in *S. aureus*.

2. In figure legend 2 the strains and strain numbers are listed. It would be helpful if the authors could indicate which strains are VRE in the figure legend so the reader does not have to refer back to Table 1.

3. The authors showed that the combination of CEL and vancomycin protect from lethal infection in a *G. melonella* model. It would be interesting to know if this combination protects in a mammalian model system.

4. While perhaps difficult, it would be nice if the authors could quantify the differences in cell length.

5. The authors assert that CEL targets FtsZ and this allows for synergy with vancomycin. Do other known inhibitors of FtsZ also promote synergy with vancomycin? Does ftsZ mutation in VRE confer susceptibility to vancomycin?

6. One drawback is the correlation with CEL binding FtsZ and suppressing GTPase activity as the primary mechanism of action in inducing vancomycin susceptibility. Would CEL still synergize with a VRE strain harboring a GTPase deficient FtsZ?

Preparing Revision Guidelines

- Point-by-point responses to the issues raised by the reviewers in a file named "Response to Reviewers," NOT IN YOUR COVER LETTER.

- Upload a compare copy of the manuscript (without figures) as a "Marked-Up Manuscript" file.
- Each figure must be uploaded as a separate file, and any multipanel figures must be assembled into one file.
- Manuscript: A .DOC version of the revised manuscript
- Figures: Editable, high-resolution, individual figure files are required at revision, TIFF or EPS files are preferred

Please return the manuscript within 60 days; if you cannot complete the modification within this time period, please contact me. If you do not wish to modify the manuscript and prefer to submit it to another journal, please notify me of your decision immediately so that the manuscript may be formally withdrawn from consideration by Microbiology Spectrum.

The manuscript by Lu et al. examines the synergistic effect of celastrol with vancomycin against *Enterococcus faecalis* and *Enterococcus faecium*. Enterococci are serious health concern and the emergence of vancomycin resistant strains highlights the need for new treatment strategies. The research presented here shows that celastrol (CEL) displays anti-biofilm activity and synergizes with vancomycin to kill Enterococci including vancomycin-resistant strains (VRE) *in vitro*. Additionally, using a *Galleria melonella* larvae infection model, Liu et al. show that the combination of CEL and vancomycin protected larvae from lethal infection of VRE compared to either compound alone. Lui et al. observed elongated cells in SEM micrographs and hypothesized that the filamenting temperature-sensitive mutant Z (FtsZ) might be the target of CEL. They demonstrated by surface plasmon resonance analysis the CEL does have a strong binding ability with FtsZ and suppressed the GTPase activity that is essential for regulating cell division. Overall, the work presented here is very solid. A major highlight is the exciting finding that CEL can render VRE strains sensitive to vancomycin both *in-vitro* and *in-vivo*.

Minor comments:

1. The impact of the work shown is somewhat diminished by the fact that the same phenomena reported here has previously been reported in *S. aureus*.
2. In figure legend 2 the strains and strain numbers are listed. It would be helpful if the authors could indicate which strains are VRE in the figure legend so the reader does not have to refer back to Table 1.
3. The authors showed that the combination of CLE and vancomycin protect from lethal infection in a *G. melonella* model. It would be interesting to know if this combination protects in a mammalian model system.
4. While perhaps difficult, it would be nice if the authors could quantify the differences in cell length.
5. The authors assert that CEL targets FtsZ and this allows for synergy with vancomycin. Do other known inhibitors of FtsZ also promote synergy with vancomycin? Does *ftsZ* mutation in VRE confer susceptibility to vancomycin?
6. One drawback is the correlation with CEL binding FtsZ and suppressing GTPase activity as the primary mechanism of action in inducing vancomycin susceptibility. Would CEL still synergize with a VRE strain harboring a GTPase deficient FtsZ?

November 20th, 2022

Dr. Conlon

Editor, Microbiology Spectrum,

Re: Manuscript Spectrum03699-22 entitled “Antibacterial activity of celastrol and its synergistic effect with vancomycin against enterococci in vitro and in vivo”

Dear Dr. Conlon,

Thank you and the reviewers for the thoughtful and careful consideration of our manuscript. We appreciate the favorable review and potential acceptability of the revised paper. We have responded to the comments in detail and revised the manuscript accordingly (highlighted with yellow color). We believe that the current revision addresses all the critique points. Below is a point-by-point response to both reviewers.

Reviewer #1

1. While I found the experimental results compelling, the writing could be significantly improved. This would not only help readability but also the impact that these exciting results will have with their audience.

Response: Thanks for pointing this out. We got help from Carol Mita Ann of Harvard Medical School for revision, went over the manuscript and made changes where necessary.

2. In general, the use of color in the Figures 2-4 would help the reader.

Response: Thanks for your suggestion. We made changes accordingly and found it's very helpful to make it easy for the readers to understand the figures by using color. The changed figures as follows:

Fig 2. Activity of CEL at different concentrations in a time-kill analysis against 6 VRE strains

Fig.3. Concentration-dependent activity on biofilm eradication of CEL, vancomycin, and ampicillin.

Fig. 4 Sub-MIC levels of CEL and vancomycin alone and in combination against VRE strains.

3. Related to the "time-kill" assay, I would like to note that VAN is generally cited as

having a bacteriostatic effect on enterococci and the mechanism of FtsZ inhibition is also suggestive to slow/impair division rather than have an active mechanism of killing (such as membrane disruption). Moreover, the time-dependence of the CFU reduction and the fact that in all cases a substantial number of bacterial cells were recovered after 24 h, the use of the term bacteriocidal may be an overstatement. There does not appear to be a rapid mode of action but rather, I anticipate, a multifactorial process that ultimately leads to cell death. While the authors define bacteriocidal as a reduction by 3log10 units (99.9% reduction) the term is a bit confounding (in my opinion).

For the combination treatments CEL + VAN, I would also recommend repeating at least one assay (with one-two strains a VSE and VRE, perhaps) at concentrations closer to the MIC values. This would presumably allow readers to observe a more significant reduction in cell viability over time and better corroborate the aforementioned bacteriocidal claims in the manuscript.

Response: Thanks for your suggestion. In our research, we noticed that CEL had bacteriostatic activity, instead of bactericidal activity as you reminded. We made changes accordingly as follows: Kinetics of antibacterial activity of CEL were assessed by time-kill assay according to the method described by CLSI (CLSI, 1999). For all strains tested, significant antibacterial activity was noted at 24 h post inoculation at the concentration of 4 MIC, namely 16, 8, 16, 8, 8 and 8 µg/mL for *E. faecalis* ATCC 700802, ATCC 51575, 09-9, *E. faecium* ATCC 700221, ATCC 51559 and 8-30, respectively.

Please see Line 98 and 297 in the revised version.

For the combination treatments CEL + VAN, we repeated the time-kill assay as the reviewer suggested, with *E. faecalis* ATCC 700802. The results are as follows:

The result showed that at concentrations closer to the MIC values, the combinations of CEL and VAN still exhibited bacteriostatic activity rather than bactericidal activity.

4. The reported FICI are limited by the VAN MIC data. For strains with MICs >256 µg/mL, I do not expect the authors to determine actual values; however, the reported FICI should reflect this. In fact, many of the FICIs are lower than the reported value (e.g. for ATCC51559 the FICI is <0.25). I recommend noting this in the table near the FICI heading.

Response: Thanks for your suggestion. We agree with the editor and made changes accordingly.

Please see Line 552 in the current version.

5. The authors should relate the concentrations used in the Galleria model to those obtained from in vitro MIC determinations.

Response: Thanks for pointing this out. The MIC of CEL to most strain tested are 2 or 4 µg/mL. The weight of the *Galleria mellonella* larvae we used in this study is 0.25 ~ 0.3 g. The hemolymph percentage is about 70% of the larvae weight (According to Astvad KMT, Meletiadis J, Whalley S, Arendrup MC. Fluconazole Pharmacokinetics in *Galleria mellonella* Larvae and Performance Evaluation of a Bioassay Compared to Liquid Chromatography-Tandem Mass Spectrometry for Hemolymph Specimens. *Antimicrob Agents Chemother.* 2017 Sep 22;61(10):e00895-17). The dose of CEL we used *in vivo* is 0.25 mg/kg, equivalent to 0.25

$\text{mg/kg} \times (0.3 \times 10^{-3} \text{kg}) / (0.3 \text{mL} \times 70\%) = 0.357 \text{ } \mu\text{g/mL}$, which is close to the in vitro MIC of CEL used in combination with vancomycin (0.25 $\mu\text{g/mL}$).

Please see Line 205-208 in the current version.

Reviewer #2

1. The impact of the work shown is somewhat diminished by the fact that the same phenomena reported here has previously been reported in S. aureus.

Response: Thanks for the suggestion. Although it has been reported that CEL displayed anti-staphylococcal activity, in-depth research focused on its anti-enterococcal activity has not been seen, especially its synergic effect with vancomycin. VRE is the top ranked gram-positive bacteria on the WHO high priority list of pathogens that require urgent intervention, therefore, new strategies would provide promising therapeutic options in combating this kind of pathogens.

2. In figure legend 2 the strains and strain numbers are listed. It would be helpful if the authors could indicate which strains are VRE in the figure legend so the reader does not have to refer back to Table 1.

Response: Thanks for the suggestion, and we agree with the editor. All of the six strains are VRE, and the corresponding change has been made in the revised manuscript, please see Line 503-504 in the revised version.

3. The authors showed that the combination of CLE and vancomycin protect from lethal infection in a G. melonella model. It would be interesting to know if this combination protects in a mammalian model system.

Response: Thanks for the suggestion. We did the pilot study of the in vivo efficacy of CEL on a mouse infection model not long ago, with 3 mice per group. Mice were infected peritoneally with 1×10^7 CFU cells of *E. faecalis* strain ATCC 700802. Groups of three mice were given vancomycin (4 mg/kg), CEL (4 mg/kg), or combination of vancomycin and CEL 2h post infection. The mice were sacrificed and CFU recovered

from mouse organs including hearts, livers, spleens, lungs and kidneys were enumerated. The results are as follows:

CFU recovered from *Enterococcus faecalis* ATCC 700802-infected mouse organs

It looked promising. We are considering to do further study in the future to optimize the doses of bacterial inocula and the dose of CEL and vancomycin used *in vivo*. Besides, more mice will be used in each group to get reliable results.

4. While perhaps difficult, it would be nice if the authors could quantify the differences in cell length.

Response: As suggested, the estimated cell length of the bacteria after the treatment of CEL or not was added. Please see Lines 149-153 and 155-156 in the revised version.

5. The authors assert that CEL targets *FtsZ* and this allows for synergy with vancomycin. Do other known inhibitors of *FtsZ* also promote synergy with vancomycin?

Does ftsZ mutation in VRE confer susceptibility to vancomycin?

Response: Thanks for the suggestion. As reported, synergistic actions of some antibiotics and FtsZ inhibitors were observed. The third-generation oral cephalosporin cefdinir and TXA709, an FtsZ-targeting prodrug against methicillin-resistant *Staphylococcus aureus* (MRSA) (Kaul M. et al., 2016), exhibited synergy *in vitro* and *in vivo*. A quinoline-based FtsZ inhibitor combined with β -lactam antibiotics shows strong synergistic antimicrobial activities against antibiotic-resistant strains of *S. aureus* (Fang Z. et al., 2018). Doxorubicin, a U.S. FDA approved drug displayed strong interaction with FtsZ (Tripathy S. et al., 2018), was used as a positive control of FtsZ inhibitor in our study. Doxorubicin exhibited synergy with vancomycin, and the MIC of vancomycin against *E. faecalis* ATCC 700802 decreased significantly from 16 $\mu\text{g}/\text{mL}$ to 4 $\mu\text{g}/\text{mL}$ when been used in combination with Doxorubicin, with an FIC index of 0.375.

FtsZ is a cell division essential gene conserved in most bacteria, therefore, it is difficult to construct or select the mutant strains. To the best of our knowledge, the reported *ftsZ* mutations in gram positive bacteria were all screened resistant mutants at concentrations higher than MIC. We tried this assay, however no mutant survived. Instead, the *ftsZ*-knockdown *Bacillus subtilis* (*B. subtilis*) strain by CRISPR interference (CRISPRi) from *B. subtilis* CRISPRi Essential Gene Knockdown Library was employed in the present study. Gene knockdown was induced by xylose (1%, freshly prepared) in the CRISPRi strain. The no sgRNA wild type strain, an unrelated gene *ispH* knockdown strain, and the uninduced *ftsZ* CRISPRi strain (no xylose added) were used as controls. As shown in Table 3, lowering the levels of *ftsZ* could increase the susceptibility of vancomycin, with the MIC decreasing from 0.5 to 0.125 $\mu\text{g}/\text{mL}$. While both MICs of vancomycin and CEL in the non-induced CRISPRi strain and the unrelated gene *ispH* knockdown strain were the same as the no sgRNA wild-type parental strain.

Please see Line 218-228 and 233-243 in the current version.

Table 3 MICs of CEL and vancomycin against FtsZ knockdown strain

MIC ($\mu\text{g}/\text{mL}$)	WT	ftsZ knockdown	ftsZ CRISPRi	ispH knockdown
----	-----------------------	---------------------	-----------------------

		(xylose induced)	(uninduced)	(xylose induced)
Van	0.5	0.125	0.5	0.5
CEL	0.5	0.25	0.5	0.5

6. One drawback is the correlation with CEL binding FtsZ and suppressing GTPase activity as the primary mechanism of action in inducing vancomycin susceptibility.

Would CEL still synergize with a VRE strain harboring a GTPase deficient FtsZ?

Response: Thanks for the suggestion. As mentioned above, *ftsZ* mutant VRE strain could not be obtained under the current experiment conditions. The *FtsZ*-knockdown *Bacillus subtilis* (*B. subtilis*) strain by CRISPR interference (CRISPRi) from *B. subtilis* CRISPRi Essential Gene Knockdown Library was employed to investigate the antibacterial effect of CEL on GTPase deficient strain. Gene knockdown was induced by xylose (1%, freshly prepared) in the CRISPRi strain. The no sgRNA wild type strain, an unrelated gene *ispH* knockdown strain, and the uninduced *ftsZ* CRISPRi strain (no xylose added) were used as controls. As shown in Table 3, the decreased MIC of CEL was observed in the strain with lower expression level of *ftsZ* compared with the wild-type counterpart. While both MICs of vancomycin and CEL in the non-induced CRISPRi strain and the unrelated gene *ispH* knockdown strain were the same as the no sgRNA wild-type parental strain. The synergistic effect of CEL and vancomycin could not be seen in *ftsZ* knockdown strain. The possible explanation could be that the deficient GTPase activity (due to gene knockdown) exhibited an equivalent effect produced by a cell division inhibitor (e.g. CEL). Therefore, CEL could not further exhibit reinforcement to the activity of vancomycin in the *ftsZ* knockdown strain, which reconfirmed the action of CEL on the FtsZ protein.

Please see Line 233-251 in the new version.

The manuscript was also formatted according to Microbiology Spectrum guidelines.

In addition, since Wei Guo and Zhimeng Zhang performed part of the experiments

during the manuscript revision process, we would like to add them as co-authors, which have got agreement from all the authors.

Again, we appreciate the helpful suggestions from you and the reviewers, and feel we have addressed each of them in a positive way. Thus, we feel the manuscript is improved and trust that it is now suitable for acceptance. Thank you for your consideration and we look forward to hearing from you.

Best Regards,

Sincerely,

Jing Pang, PhD.

Institute of Medicinal Biotechnology

Chinese Academy of Medical Sciences & Peking Union Medical College

Beijing 100050, China

December 2, 2022

Dr. Jing Pang
Institute of Medicinal Biotechnology, PUMC
Beijing Key Laboratory of Antimicrobial Agents
1 Tiantanxili
Beijing
China

Re: Spectrum03699-22R1 (Antibacterial activity of an FtsZ inhibitor celastrol and its synergistic effect with vancomycin against enterococci in vitro and in vivo)

Dear Dr. Jing Pang:

Thank you for submitting your manuscript to Microbiology Spectrum. As you will see your paper is very close to acceptance. Please modify the manuscript along the lines I have recommended. As these revisions are quite minor, I expect that you should be able to turn in the revised paper in less than 30 days, if not sooner. You will find the reviewers' comments below. Specifically, reviewer 1 recommends integrating the new in vitro data displayed in the response to reviewers document.

When submitting the revised version of your paper, please provide (1) point-by-point responses to the issues raised by the reviewers as file type "Response to Reviewers," not in your cover letter, and (2) a PDF file that indicates the changes from the original submission (by highlighting or underlining the changes) as file type "Marked Up Manuscript - For Review Only". Please use this link to submit your revised manuscript. Detailed instructions on submitting your revised paper are below.

Link Not Available

Sincerely,

Brian Conlon

Reviewer comments:

Reviewer #1 (Comments for the Author):

I thank the authors for their responses to my suggestions and concerns; however, I am puzzled as to the reason for omission of newly obtained data from the manuscript. In response to comment 3, the authors performed the requested experiment but did not update Fig 4. While I understand the reluctance to include the preliminary mouse study, the in vitro experiment showing additional concentrations/combinations of CEL and VAN could be easily included in the manuscript. In fact, this was requested for the broader audience. I am recommending that it is integrated into Figure 4 - perhaps just the higher concentrations inset into Fig 4A - or that it is provided to readers as a supplemental figure.

Minor comment

The image resolution of Figure 7 (right hand side) and 8 are low.

Reviewer #2 (Comments for the Author):

The authors have adequately addressed the previous reviewer comments.

Preparing Revision Guidelines

Please return the manuscript within 60 days; if you cannot complete the modification within this time period, please contact me. If you do not wish to modify the manuscript and prefer to submit it to another journal, please notify me of your decision immediately so that the manuscript may be formally withdrawn from consideration by Microbiology Spectrum.

The authors have adequately addressed the previous reviewer comments. The new version of the manuscript is much improved and as such I do not believe further revisions are needed.

December 3rd, 2022

Dr. Conlon

Editor, Microbiology Spectrum,

Re: Manuscript Spectrum03699-22R1 entitled “Antibacterial activity of celastrol and its synergistic effect with vancomycin against enterococci in vitro and in vivo”

Dear Dr. Conlon,

Thank you and the reviewers for the thoughtful and careful consideration of our manuscript. We appreciate the favorable review and potential acceptability of the revised paper. We have responded to the comments in detail and revised the manuscript accordingly (highlighted with green color). We believe that the current revision addresses all the critique points. Below is a point-by-point response to both reviewers.

Reviewer #1

1. I am puzzled as to the reason for omission of newly obtained data from the manuscript. In response to comment 3, the authors performed the requested experiment but did not update Fig 4. While I understand the reluctance to include the preliminary mouse study, the in vitro experiment showing additional concentrations/combinations of CEL and VAN could be easily included in the manuscript. In fact, this was requested for the broader audience. I am recommending that it is integrated into Figure 4 - perhaps just the higher concentrations inset into Fig 4A - or that it is provided to readers as a supplemental figure.

Response: Thanks for the suggestion, and we agree with the reviewer. We made changes accordingly. Since the integration would bring too much curves to the Fig. 4A, which might be difficult for the readers to figure them out, we provide the new figure in the supplemental material separately.

Please see Line 131-134 and Line 549-551 (Fig. S1) in the revised version.

2. *The image resolution of Figure 7 (right hand side) and 8 are low.*

Response: Thanks for the suggestion, and we uploaded the new Figure 7 and Figure 8 with a higher resolution.

Again, we appreciate the helpful suggestions from you and the reviewers, and feel we have addressed each of them in a positive way. Thus, we feel the manuscript is improved and trust that it is now suitable for acceptance. Thank you for your consideration and we look forward to hearing from you.

Best Regards,

Sincerely,

Jing Pang, PhD.

Institute of Medicinal Biotechnology

Chinese Academy of Medical Sciences & Peking Union Medical College

Beijing 100050, China

December 9, 2022

Dr. Jing Pang
Institute of Medicinal Biotechnology, PUMC
Beijing Key Laboratory of Antimicrobial Agents
1 Tiantanxili
Beijing
China

Re: Spectrum03699-22R2 (Antibacterial activity of an FtsZ inhibitor celastrol and its synergistic effect with vancomycin against enterococci in vitro and in vivo)

Dear Dr. Jing Pang:

Your manuscript has been accepted, and I am forwarding it to the ASM Journals Department for publication. You will be notified when your proofs are ready to be viewed.

Sincerely,

Brian Conlon
Editor, Microbiology Spectrum

Journals Department
Supplemental file 1: Accept